# Selective agonist of TRPML2 reveals direct role in chemokine release from innate immune cells

Eva Plesch[1†], Cheng-Chang Chen[1†], Elisabeth Butz[1†], Anna Scotto Rosato[2], Einar K Krogsaeter[3], Hua Yinan[4], Karin Bartel[1], Marco Keller[1], Dina Robaa[5], Daniel Teupser[6], Lesca M Holdt[6], Angelika M Vollmar[1], Wolfgang Sippl[5], Rosa Puertollano[4], Diego Medina[2], Martin Biel[1], Christian Wahl-Schott[7]*, Franz Bracher[1]*, Christian Grimm[3]*

[1]Department of Pharmacy, Center for Drug Research and Center for Integrated Protein Science Munich, Ludwig Maximilian University of Munich, Munich, Germany; [2]Telethon Institute of Genetics and Medicine, Naples, Italy; [3]Department of Pharmacology and Toxicology, Medical Faculty, Ludwig Maximilian University of Munich, Munich, Germany; [4]Cell Biology and Physiology Center, National Heart, Lung, and Blood Institute, National Institutes of Health, Bethesda, United States; [5]Department of Pharmaceutical Chemistry, Institute of Pharmacy, Martin Luther University of Halle-Wittenberg, Halle, Germany; [6]Institute of Laboratory Medicine, University Hospital Munich, Munich, Germany; [7]Institute for Neurophysiology, Hannover Medical School, Hannover, Germany

*For correspondence:
christian.wahl@cup.uni-muenchen.de (CW-S);
franz.bracher@cup.uni-muenchen.de (FB);
chgrph@cup.uni-muenchen.de (CG)

†These authors contributed equally to this work

Competing interests: The authors declare that no competing interests exist.

**Abstract** Cytokines and chemokines are produced and secreted by a broad range of immune cells including macrophages. Remarkably, little is known about how these inflammatory mediators are released from the various immune cells. Here, the endolysosomal cation channel TRPML2 is shown to play a direct role in chemokine trafficking and secretion from murine macrophages. To demonstrate acute and direct involvement of TRPML2 in these processes, the first isoform-selective TRPML2 channel agonist was generated, ML2-SA1. ML2-SA1 was not only found to directly stimulate release of the chemokine CCL2 from macrophages but also to stimulate macrophage migration, thus mimicking CCL2 function. Endogenous TRPML2 is expressed in early/recycling endosomes as demonstrated by endolysosomal patch-clamp experimentation and ML2-SA1 promotes trafficking through early/recycling endosomes, suggesting CCL2 being transported and secreted via this pathway. These data provide a direct link between TRPML2 activation, CCL2 release and stimulation of macrophage migration in the innate immune response.
DOI: https://doi.org/10.7554/eLife.39720.001

## Introduction

Cytokines/chemokines are released from a wide range of immune cells such as macrophages, B- and T-lymphocytes, neutrophils, mast cells and dendritic cells. They are essential for intercellular communication in both innate and adaptive immunity. Remarkably, our knowledge of the function of cytokines/chemokines in immunity is much more advanced than our knowledge about how they are packaged and secreted from immune cells. Understanding how innate immune cells release cytokines/chemokines is important, as these factors are indispensable for communication between immune but also with non-immune cells to coordinate inflammatory responses (*Lacy and Stow, 2011*). Importantly, secretion pathways vary between different cell types. Macrophages for example

lack typical secretory granules (*Lacy and Stow, 2011*). Thus, macrophage cytokine/chemokine release is mediated either by direct transport to the cell surface from the trans-Golgi network (TGN) (e.g. IL-10), by transport via recycling endosomes (RE) to the cell surface (e.g. TNF-α, IL-6, IL-10) (*Manderson et al., 2007*; *Murray and Stow, 2014*), or via late endosomes/lysosomes (LE/LY), for example IL-1β (*Andrei et al., 1999*; *Lopez-Castejon and Brough, 2011*).

We show here that the endolysosomal calcium-permeable cation channel TRPML2 plays a direct role in chemokine secretion, thereby modulating the inflammatory response. Expression of TRPML2 in different immune cells and tissues has been demonstrated by several groups (*Cuajungco et al., 2016*; *Valadez and Cuajungco, 2015*; *García-Añoveros and Wiwatpanit, 2014*; *Sun et al., 2015*). On the subcellular level, TRPML2 has been shown to be expressed primarily on RE and LE/LY by immunocytochemistry experiments (*Sun et al., 2015*; *Venkatachalam et al., 2006*; *Karacsonyi et al., 2007*). However, functional expression of TRPML2 in different intracellular vesicles and organelles has not been confirmed yet by direct and selective patch-clamp analysis, that is patch-clamping of RE, EE (early endosomes), LE/LY, or other endolysosomal vesicles. Furthermore, it remains unclear whether direct and selective stimulation of TRPML2 leads to an increase in cytokine/chemokine release from macrophages, and which intracellular trafficking pathways mediate the release of these cytokines/chemokines.

One important impediment for the investigation of different endogenous TRPML-like currents and their functional impact on secretion, endolysosomal trafficking, or autophagy, is the lack of iso-form-selective agonists. Development of such agonists would allow demonstration of the TRPML iso-form-specific contribution towards observed phenomena, for example chemokine secretion. Currently available TRPML channel agonists belong to different chemotypes, including benzenesulfo-namides (e.g. SN-1- or SF-21-type), thiophenesulfonamides (e.g. SF-22-type, including MK6-83), iso-indolediones (e.g. SF-51-type, including ML-SA1), isoxazolines (e.g. SN-2-type) and others (*Grimm et al., 2010*; *Yamaguchi and Muallem, 2010*; *Grimm et al., 2012b*; *Shen et al., 2012*). Effi-cacy, potency and selectivity of these compounds can vary between species. Furthermore, none of the currently available TRPML agonists is selective for TRPML1 or TRPML2. ML-SA1 for example acti-vates TRPML1 and TRPML3 in mouse, while it activates all three human isoforms (*Shen et al., 2012*; *Grimm, 2016*). MK6-83 activates TRPML1 and TRPML3 in both mouse and human (*Grimm, 2016*; *Chen et al., 2014*). The putative endogenous TRPML channel activator $PI(3,5)P_2$ activates all three TRPML channel isoforms in both species and, in addition, also activates the endolysosomal cation channels TPC1 and TPC2 (*Chen et al., 2014*; *Wang et al., 2012*; *Cang et al., 2013*; *Grimm et al., 2014*). Through systematic chemical modification of known lead structures we have now generated the first isoform-selective TRPML2 channel agonist, ML2-SA1.

We demonstrate that ML2-SA1 activates TRPML2 in EE and LE/LY as well as in Rab11+ and Tf+/TfR+ (transferrin/transferrin receptor) vesicles. In macrophages, LPS (lipopolysaccharide) exposure leads to a strong upregulation of TRPML2 expression, while TRPML1 and TRPML3 expression levels remain unaffected by LPS (*Sun et al., 2015*). Importantly, activation by ML2-SA1 was not observed in macrophages without LPS treatment which express TRPML2 only at very low levels, further con-firming specificity of the compound. We also show that direct activation of TRPML2 by ML2-SA1 results in an increased release of the chemokine CCL2 from LPS-stimulated WT macrophages, while TRPML2$^{-/-}$ macrophages show no release increase, suggesting that TRPML2 channel activity is directly linked to CCL2 trafficking and secretion. We further provide evidence that CCL2 is released via the early/recycling endosomal pathway but not via LE/LY. Finally, we show that stimulation with ML2-SA1 promotes macrophage migration, one of the major physiological functions of the chemoat-tractant CCL2, one synonym of which is monocyte chemoattractant protein 1 (MCP-1).

## Results

### Development of a potent isoform-selective TRPML2 channel agonist

With the aim to further improve the characteristics of existing TRPML channel agonists, we gener-ated more than 80 novel derivatives of recently reported lead activators of TRPML channels which had been originally identified by random screening of the MLSMR small molecule library (Scripps Research Institute Molecular Screening Center) (*Grimm et al., 2010*). Here, novel derivatives of the lead compounds SN-2 and ML-SA1, a SF-51 analogue (*Grimm et al., 2010*; *Shen et al., 2012*;

*Grimm, 2016*; *Chen et al., 2014*) were evaluated for their efficacy, potency, and selectivity profiles, respectively.

We first synthesized and tested >50 chemically modified versions of the TRPML3 activator SN-2 (*Figure 1*; *Figure 1—figure supplement 1*; *Supplementary file 1*). These modifications comprise systematic variations of the substitution pattern of the aryl ring, variations of the aliphatic norbornane ring system, aromatisation of the isoxazoline to an isoxazole fragment, introduction of polar substituents, as well as replacement of the isoxazol(in)e ring by other heterocycles. Crucial steps in these syntheses were Huisgen-type 1,3-dipolar cycloaddition reactions of norbornene (for the closer analogues) and other alkenes with nitrile oxides (*Jawalekar et al., 2011*; *Huisgen, 1963*) and related 1,3-dipoles. Related aromatic isoxazole analogues were prepared via cycloaddition of nitrile oxides with ketone enolates (*Vitale and Scilimati, 2013*) or enamines (*Fos et al., 1992*). General synthesis strategies for these modifications are shown in *Figure 1A*.

Derivatives of SF-51/ML-SA1 (*Figure 1—figure supplement 2*; *Figure 1—figure supplement 3*; *Supplementary file 1*) were synthesized by combining appropriate amine building blocks (partially hydrogenated quinolines and other cyclic and open-chain analogues) with *N*-acyl spacers and imide/lactam-type residues following standard procedures (*Figure 1—figure supplement 3A*).

Following synthesis, we initially tested the compounds in HEK293 cells transiently transfected with human TRPML1, TRPML2, or TRPML3 (C-terminally fused to YFP) by using the fura-2 calcium imaging technique. When expressed in HEK293 cells, TRPML2 and TRPML3 but not TRPML1 substantially localize at the plasma membrane besides endolysosomes as described previously (*Grimm et al., 2010*), enabling standard fura-2 calcium imaging experimentation. To evaluate effects on TRPML1, a plasma membrane variant with mutated lysosomal targeting sequences in the N- and C-termini (TRPML1(NC)) was used as reported previously (*Grimm et al., 2010*).

The majority of the SN-2 and SF-51/ML-SA1 derivatives were either inactive, non-selective like ML-SA1, or selective for TRPML3 like SN-2 (*Figure 1B*; *Figure 1—figure supplement 3B*). A subset of molecules however displayed a strong preference for TRPML2: ML2-SA1 (=EVP-22), a derivative of SN-2, as well as derivatives of SF-51/ML-SA1: EVP-198, EVP-207 and EVP-209. The latter three SF-51/ML-SA1 derivatives however showed lower efficacy compared to ML2-SA1 (*Figure 1B*; *Figure 1—figure supplement 3B*).

## TRPML2 activity is detectable in EE, LE/LY as well as Rab11+ and TfR + organelles

In endolysosomal patch-clamp experiments using transiently transfected HEK293 cells, we investigated TRPML2 channel activity in wortmannin/latrunculin B (Wort./Lat.B)-enlarged EE (*Chen et al., 2017a*), in YM201636-enlarged LE/LY (*Chen et al., 2017a*), as well as in vacuolin-enlarged Rab11 + and TfR+ organelles (*Figure 2*; *Figure 2—figure supplement 1*). In LE/LY, both ML2-SA1 (*Figure 2B*; *Figure 2—figure supplement 1A*) and PI(3,5)$P_2$ (*Figure 2—figure supplement 1A*) evoked TRPML2 activation while no or very little activation was detectable for TRPML1 and TRPML3. In contrast, the latter ones were robustly activated by ML-SA1 as a positive control (*Figure 2C–E*). The time course for activation of TRPML2 in LE/LY patch-clamp experiments and the relative $Ca^{2+}$ permeability are shown in *Figure 2G* and *Figure 2—figure supplement 1B*. In addition to LE/LY, TRPML2 channel activity was also detectable in EE after stimulation with ML2-SA1 (*Figure 2H and K*). In order to patch-clamp discrete populations of vesicles involved in early/recycling endosomal trafficking, cells were transfected with fluorophore-tagged Rab11 or TfR, and enlarged with vacuolin. ML2-SA1 elicited significant currents in Rab11+ and in TfR+ vesicles (*Figure 2I–K*).

Furthermore, the effect of luminal pH on TRPML2 channel activity was evaluated (*Figure 2—figure supplement 1A and C*). TRPML2 activity (stimulated with PI(3,5)$P_2$ or with ML2-SA1) increases with increasing, that is less acidic luminal pH. This differs from TRPML1 which shows maximal activity in highly acidic luminal pH (*Chen et al., 2017a*; *Dong et al., 2010*). These findings argue TRPML2 channel function is adapted to vesicles of only slightly acidic or neutral pH such as EE/RE rather than highly acidic LE/LY. The strong colocalization between TfR or Rab11 with TRPML2 confirms an important functional role of TRPML2 in RE (*Figure 2—figure supplement 2A–B*).

In summary, ML2-SA1 was found to be a potent and efficacious activator of both hTRPML2 and mTRPML2. The calculated $EC_{50}$ values for human and mouse TRPML2 were 1.24 ± 0.12 μM and 2.38 ± 0.01 μM, respectively (*Figure 2F*; *Figure 2—figure supplement 3A*). ML2-SA1 shows high selectivity over h/mTRPML1 and h/mTRPML3 in both calcium imaging and endolysosomal patch-

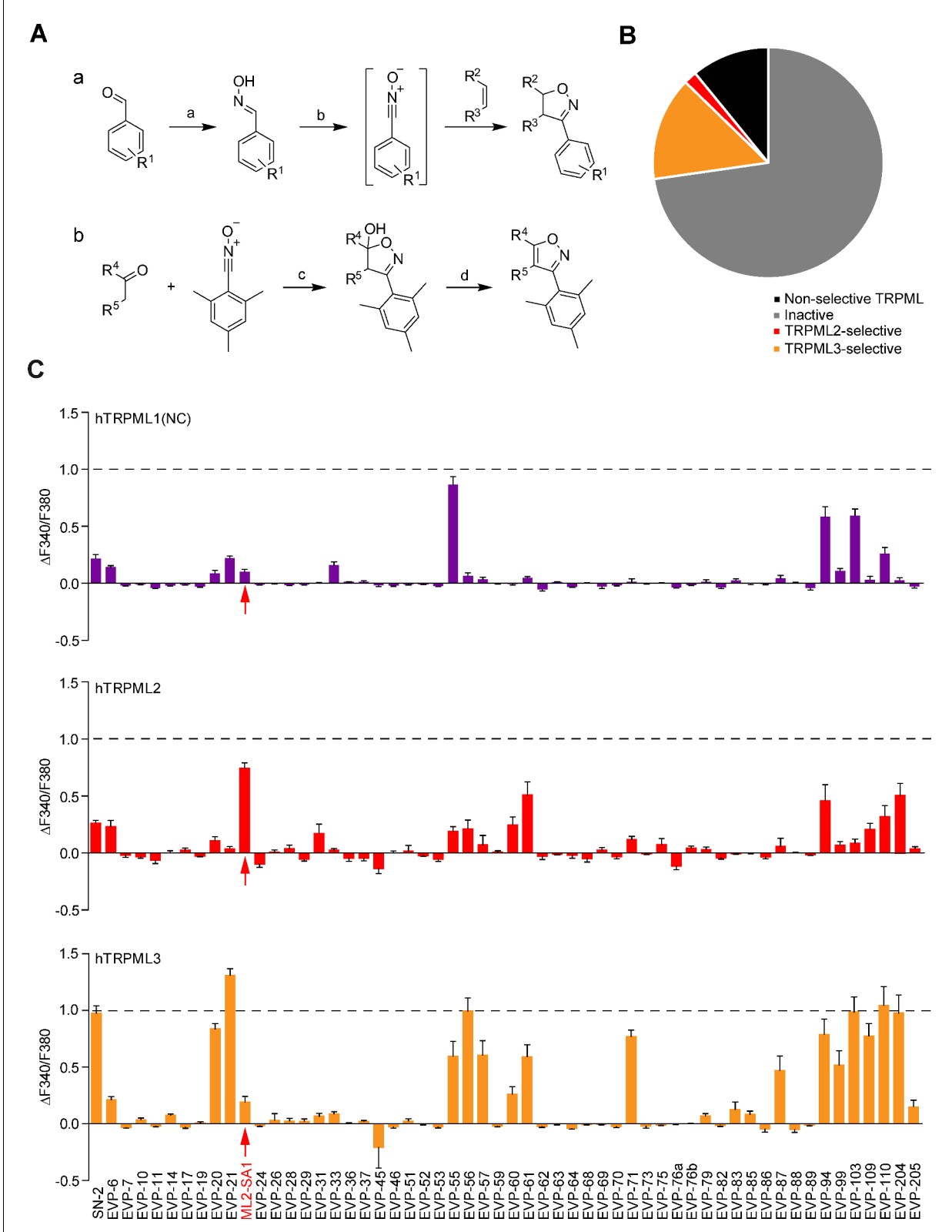

**Figure 1.** Chemical synthesis strategies and functional evaluation of SN-2 analogous compounds using calcium imaging. (**A**) Shown are synthesis strategies a and b used to generate most of the SN-2 analogous compounds shown in Suppl. **Figure 1**. $R^1$ = alkyl/ halogen/nitro/ methoxy; $R^2$ / $R^3$ = (cyclo)alkyl/phenyl/ hydroxyalkyl; $R^4$ / $R^5$=alkyl/ phenyl / (hetero)cycles; **a**) $H_2N$-OH • HCl (1.5 eq.), NaOH (3 eq.), $H_2O$:EtOH (1:1), 0°C - rt, 18 hr; **b**) PIFA (1.2 eq.), alkene (1.5 eq.), $H_2O$:MeOH (1:2), rt, 1–24 hr; **c**) ketone (2 eq.), LDA (2 eq.), THF, −78°C, 2 hr, mesitonitrile oxide, −78°C - rt, 2–15 hr; **d**)
*Figure 1 continued on next page*

*Figure 1 continued*

Na$_2$CO$_3$, MeOH:H$_2$O (2:1), 95°C, 2 hr. (**B**) Cartoon showing schematically the fractions of inactive, non-selective TRPML activating, TRPML2-selective, and TRPML3-selective agonists (total number = 55). (**C**) Fura-2 calcium imaging results showing the effect of SN-2 and its analogues (10 µM) on hTRPML1(NC)-YFP, hTRPML2-YFP, and hTRPML3-YFP transfected HEK293 cells. Mean values normalized to basal (200 s after compound application)± SEM of up to >100 independent experiments with 3–10 cells per experiment are shown.
DOI: https://doi.org/10.7554/eLife.39720.002

The following figure supplements are available for figure 1:

**Figure supplement 1.** Structures of SN-2 analogues.
DOI: https://doi.org/10.7554/eLife.39720.003
**Figure supplement 2.** Structures of SF-51/ML-SA1 analogues.
DOI: https://doi.org/10.7554/eLife.39720.004
**Figure supplement 3.** Chemical synthesis strategies and functional evaluation of SF-51/ML-SA1-analogous compounds using calcium imaging.
DOI: https://doi.org/10.7554/eLife.39720.005

clamp experiments and it does not activate TPC1 nor TPC2 (*Figure 1*; *Figure 2*; *Figure 2—figure supplement 3B–F*).

## Molecular modeling of ML2-SA1 binding

Several recent papers have provided in-depth information on the structures of TRPML1 and TRPML3 channels (*Schmiege et al., 2017*; *Chen et al., 2017b*; *Hirschi et al., 2017*). *Schmiege et al., 2017* found that a hydrophobic cavity created by I468 and F465 of PH1 (pore helix 1), F428, C429, V432 and Y436 of S5, F505 and F513 of S6, and Y499 and Y507 of S6 in the neighboring subunit, tightly accommodates ML-SA1 (*Figure 3A*). In a molecular modeling approach using these recently published structures of TRPML1 and TRPML3 as a basis, we simulated the binding of ML-SA1 as well as ML2-SA1 to hTRPML1 and hTRPML2 (*Figure 3*; *Figure 3—figure supplement 1*). Complete 3D models of the open conformation of hTRPML1 and hTRPML2 were constructed and used for ligand docking analysis. Amino acids differing between hTRPML1 and hTRPML2 are colored green (*Figure 3B–D*). Based on this model, ML2-SA1 (both enantiomers are described, one in *Figure 3—figure supplement 1*) is predicted to bind to the same binding pocket as ML-SA1 as observed in the cryo-EM structure of hTRPML1 (*Figure 3A–B*). Six amino acids (A422, A424, G425, A453, V460, and I498) in this pocket are unique to hTRPML2 (highlighted in green; *Figure 3C–D*). The orientation of ML2-SA1 in the binding pocket of hTRPML2 with the highest docking score is shown in *Figure 3C*. The dichlorophenyl ring shows favorable π-stacking interaction with F502 whereas the polar isoxazole ring is located near the side chain OH-groups of Y428 and Y496. The hydrophobic norbornane ring is interacting with G425 and Y428. Other possible orientations of ML2-SA1 binding to hTRPML2 are shown in *Figure 3—figure supplement 1C–D*). The observed binding mode of ML2-SA1 at hTRPML1 is different and appears to be energetically less favorable compared to hTRPML2 due to the observed amino acid substitutions in the predicted binding cavity (*Figure 3D*). We subsequently replaced each of the six amino acids that are unique to the predicted hTRPML2 binding pocket with the respective amino acids of hTRPML1. We analysed these mutant isoforms first in calcium imaging experiments where we found the strongest reduction of the ML2-SA1 effect in G425A (*Figure 3E*). In the next step, we performed endolysosomal patch-clamp experiments with this mutant. Mutation of G425 to alanine was found to selectively abrogate the effect of ML2-SA1, while ML-SA1 was still able to activate G425A to a degree not significantly different from WT (*Figure 3E–F*). G425 is close to the norbornane ring of ML2-SA1 (minimum distance 3.6 Å) docked to hTRPML2 and substitution to alanine is unfavorable for this binding mode (*Figure 3C*). The experimental data corroborate binding of ML2-SA1 to the ML-SA1 binding pocket and confirm a critical role of G425 in mediating ML2-SA1 selectivity.

## Effect of ML2-SA1 on endogenous TRPML2 channel activity in organelles isolated from LPS-stimulated macrophages

In macrophages significant TRPML2 channel expression is found only after stimulation with LPS, as demonstrated previously by qRT-PCR and western blot analysis (*Sun et al., 2015*). We confirmed this finding by qRT-PCR and endolysosomal patch-clamping, revealing that only after several hours of LPS treatment, robust endogenous TRPML2 channel expression and activity were detectable

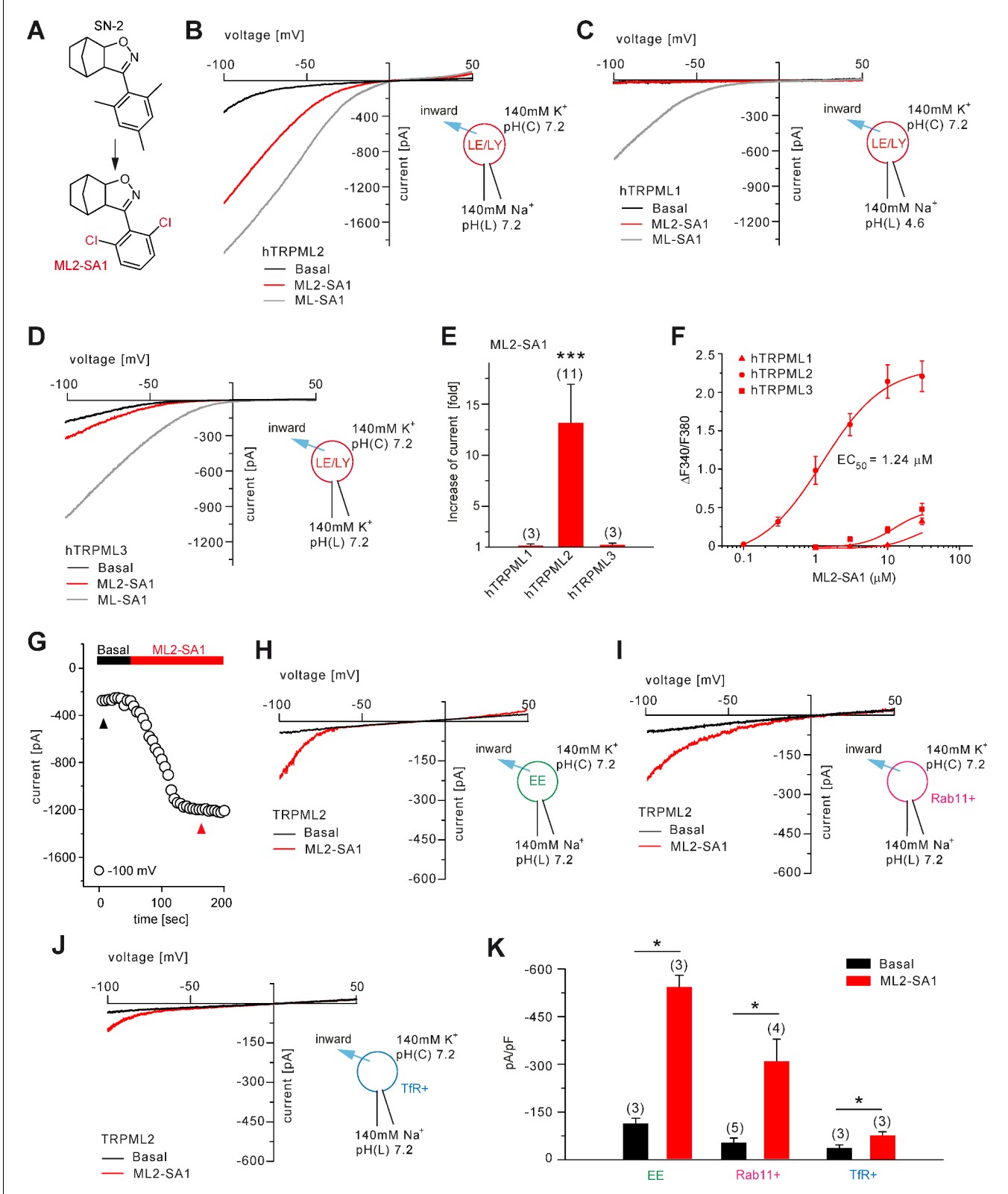

**Figure 2.** Effect of ML2-SA1 on TRPML channels. (**A**) Cartoon depicting chemical structures of SN-2 and ML2-SA1. (**B**) Representative ML2-SA1 or ML-SA1 (10 μM) elicited currents from YM201636-enlarged LE/LY isolated from hTRPML2 expressing HEK293 cells. (**C–D**) Representative ML2-SA1 or ML-SA1 (10 μM) elicited currents from YM201636-enlarged LE/LY isolated from hTRPML1 or hTRPML3 expressing HEK293 cells. (**E**) Statistical summary of ML2-SA1 data as shown in B-D as fold increase compared to the respective basal currents in LE/LY. Shown are mean values ± SEM at −100 mV of n

*Figure 2 continued on next page*

*Figure 2 continued*

independent experiments as indicated, each. (F) Dose-response curves obtained from fura-2 calcium imaging experiments with hTRPML1(NC), hTRPML2, and hTRPML3 expressed in HEK293 cells and elicited with ML2-SA1 at varying concentrations. The calculated $EC_{50}$ value for hTRPML2 is: $1.24 \pm 0.12$ µM (mean ± SEM). (G) Time course of TRPML2 activation by ML2-SA1 taken from experiments as shown in B. Black and red arrows indicate time points for basal and ML2-SA1 induced TRPML2 activity that were used for the IV relationship. (H–J) Representative basal and ML2-SA1 (10 µM) elicited currents from Wort./Lat.B-enlarged EE, from vacuolin-enlarged Rab11+, or form TfR+ vesicles isolated from hTRPML2 expressing HEK293 cells. (K) Statistical summary of data as shown in G-I. * indicates p<0.05, ** indicates p<0.01, *** indicates p<0.001, *Figure 2E*, one-way ANOVA test followed by Tukey's post-hoc test, *Figure 2J*, paired t-test.

DOI: https://doi.org/10.7554/eLife.39720.006

The following figure supplements are available for figure 2:

**Figure supplement 1.** Effect of ML2-SA1 on TRPML2 under different pH conditions.

DOI: https://doi.org/10.7554/eLife.39720.007

**Figure supplement 2.** Co-transfection of HEK293 cells with fluorescently labelled TRPML2 and vesicle-specific markers of the endolysosomal system.

DOI: https://doi.org/10.7554/eLife.39720.008

**Figure supplement 3.** DRC of ML2-SA1 effect on mTRPML2, effects of ML2-SA1, SN2, and ML-SA1 on mTRPML channel isoforms, and cytotoxicity of ML2-SA1.

DOI: https://doi.org/10.7554/eLife.39720.009

(*Figure 4A–G*). In LPS-stimulated bone marrow-derived macrophages (BMDMΦ) ML2-SA1-induced currents were detectable in Tf-Alexa555 loaded, vacuolin-enlarged vesicles, while no significant TRPML2 channel activity could be detected in non-LPS stimulated BMDMΦ Tf+ vesicles (*Figure 4A–B*). Currents measured in BMDMΦ LE/LY with ML2-SA1 after LPS-stimulation were smaller than currents measured in Tf+ loaded vesicles (*Figure 4C–D*). In contrast, in LE/LY isolated from alveolar macrophages (AMΦ), TRPML2 currents elicited with ML2-SA1 were larger on average than in BMDMΦ (*Figure 4E–F*). These data confirm that ML2-SA1 elicits robust TRPML2 currents in endogenously expressing cells.

## Effect of selective TRPML2 activation on CCL2 secretion

To evaluate effects of the novel TRPML2 channel agonist on chemokine secretion from macrophages, we performed experiments based on the results recently provided by *Sun et al. (2015)* (*Figure 5A*). We found that incubation with ML2-SA1 significantly increased secretion of the chemokine CCL2 from BMDMΦ, both after 4 hr and 8 hr of LPS treatment (*Figure 5A*). Importantly, ML2-SA1 did not induce CCL2 secretion in unstimulated BMDMΦ. Furthermore, CCL2 secretion was severely reduced in TRPML2$^{-/-}$ BMDMΦ and ML2-SA1 showed no further increase of CCL2 secretion in the TRPML2$^{-/-}$ BMDMΦ, corroborating the specificity of the agonist (*Figure 5A*). To characterise the pathway of ML2-SA1-induced CCL2 secretion from macrophages, we performed lysosomal exocytosis and Tf trafficking experiments to distinguish between LE/LY and EE/RE as possible secretion routes. Lysosomal exocytosis experiments revealed no significant effect of ML2-SA1 on lysosomal enzyme (beta-hexosaminidase) release (*Figure 5B*). In accordance with this, ML2-SA1 application did not result in translocation of LAMP1 to the plasma membrane (*Figure 5C*), arguing against LE/LY being involved in CCL2 secretion in BMDMΦ. These findings are supported by the LE/LY environment being less favorable for TRPML2 activity as outlined above. More favorable conditions are found in EE/RE compartments (less acid to neutral pH). In line with this, ML2-SA1 application resulted in a significant enhancement of Tf trafficking and recycling through EE/RE (*Figure 5D–E*). Taken together, these data argue for a TRPML2-dependent trafficking route of CCL2 from Golgi to EE/RE (*Figure 5F*).

## ML2-SA1 promotes macrophage migration

To assess effects of ML2-SA1 on cell migration, we performed migration assays in a modified Boyden chamber setup (*Figure 6—figure supplement 1*). BMDMΦ in the presence or absence of LPS were seeded in the lower compartment of the chamber and exposed to different concentrations of ML2-SA1. LPS-stimulated, ML2-SA1 pre-treated BMDMΦ were able to significantly increase migration of untreated BMDMΦ through the transwell chamber, while LPS-stimulated BMDMΦ without ML2-SA1 pre-treatment (only DMSO) were not able to alter migration properties of untreated BMDMΦ. (*Figure 6A*). This is in accordance with the enhanced release of CCL2 by ML2-SA1, which

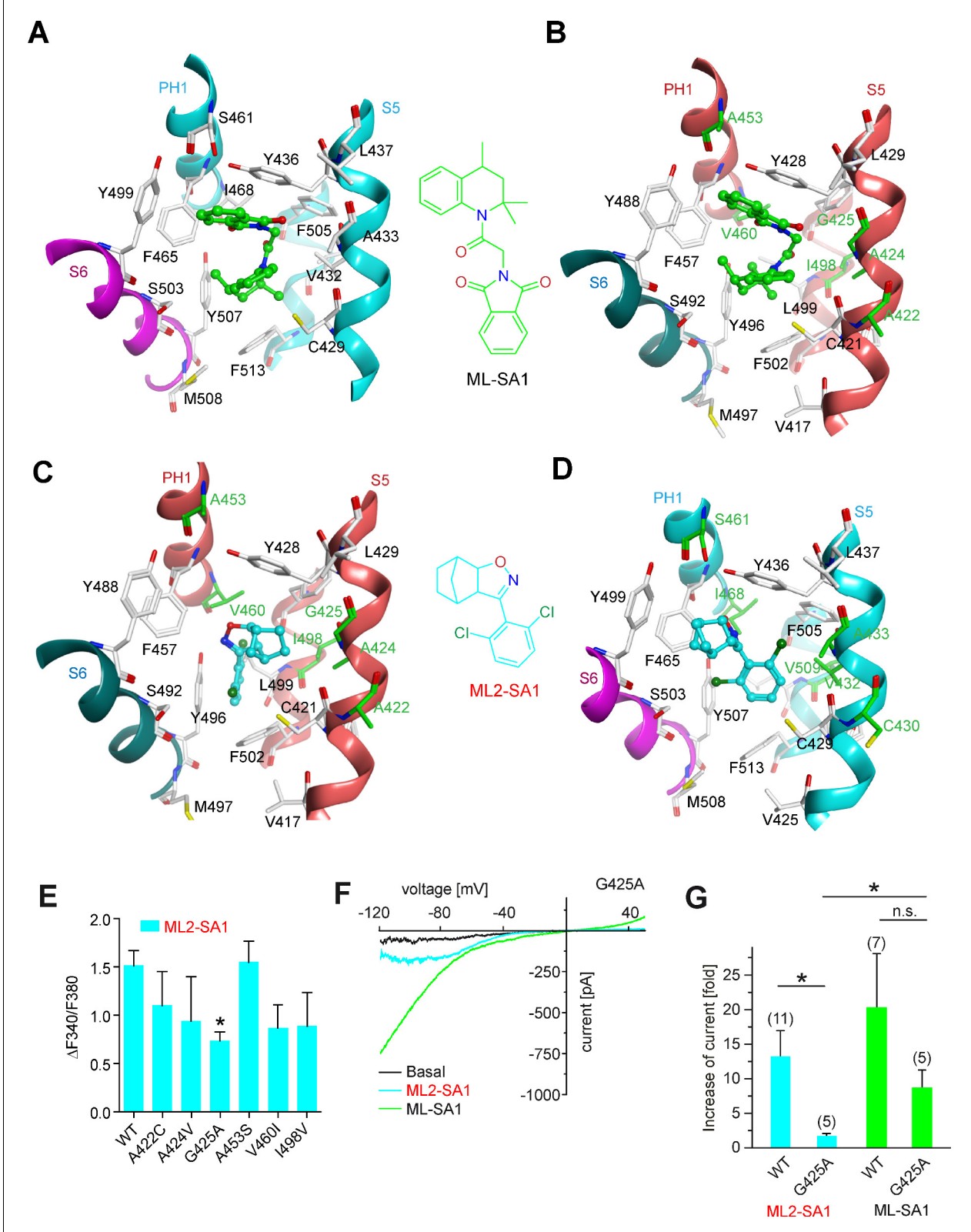

**Figure 3.** Molecular modeling of ML2-SA1 and ML-SA1 binding. (**A**) Binding mode of ML-SA1 (green colored carbon atoms) at hTRPML1, showing residues within 5 Å of ML-SA1, as observed in one of the four identical binding pockets of the cryo-EM structure (PDB ID: 5WJ9). The S6 helix of monomer A of hTRPML1 is colored magenta, the PH1 and S5 helices of monomer B are colored cyan. (**B**) Binding mode of ML-SA1 (green colored carbon atoms) at hTRPML2 (homology model generated with MODELLER) as predicted by the ligand docking. Only residues within 5 Å of ML-SA1 in
*Figure 3 continued on next page*

Figure 3 continued

one of the four identical binding pockets are displayed. The S6 helix of monomer A of hTRPML2 is colored petrol blue, the PH1 and S5 helices of monomer B are colored salmon. Amino acid residues that are different in hTRPML1 and hTRPML2 are colored green (C) Binding mode of one ML2-SA1 enantiomer (cyan colored carbon atoms; 3a*S*, 4*S*, 7*R*, 7a*S*) at hTRPML2 as predicted by ligand docking. Only residues within 5 Å of ML2-SA1 in one of the four identical binding pockets are displayed (same coloring and representation style as in *Figure 3B*). Binding of the other ML2-SA1 enantiomer (3a*R*, 4*R*, 7*S*, 7a*R*) resulted in a similar binding mode that is shown in *Figure 3—figure supplement 1B* (D) Binding mode of one ML2-SA1 enantiomer (cyan colored carbon atoms; 3a*S*, 4*S*, 7*R*, 7a*S*) at hTRPML1 as predicted by ligand docking. Only residues within 5 Å of ML2-SA1 in one of the four identical binding pockets are displayed (same coloring and representation style as in *Figure 3a*). (E) Fura-2 calcium imaging results showing the effect of ML2-SA1 (10 µM) on hTRPML2-YFP WT and mutant transfected HEK293 cells. Mean values normalized to basal (120 s after compound application)± SEM of at least three independent experiments, each. * indicates $p < 0.05$, one-way ANOVA, followed by Dunnet post-hoc test. (F) Representative ML2-SA1 or ML-SA1 (10 µM) elicited currents from YM201636-enlarged LE/LY isolated from hTRPML2(G425A) expressing HEK293 cells. (G) Statistical summary of data as shown in F as fold increase compared to the respective basal currents in LE/LY. Shown are mean values ± SEM at −100 mV of at n independent experiments as indicated. * indicates $p < 0.05$, unpaired t-test.

DOI: https://doi.org/10.7554/eLife.39720.010

The following figure supplement is available for figure 3:

**Figure supplement 1.** Additional molecular modeling of ML2-SA1 and ML-SA1 binding.

DOI: https://doi.org/10.7554/eLife.39720.011

serves as chemoattractant for untreated BMDMΦ. To exclude a chemotactic effect of the compounds themselves, we used a classical Boyden chamber setup without cells in the lower compartment. Yet, neither LPS nor ML2-SA1 alone were able to significantly enhance BMDMΦ migration, while supplementation of recombinant CCL2 led to a substantial increase in BMDMΦ migration (*Figure 6B*; *Figure 6—figure supplement 2A–B*). Overall, these data suggest that ML2-SA1 is able to induce CCL2 secretion selectively in TRPML2-expressing macrophages, thus serving as chemoattractant to recruit more macrophages.

## Discussion

We describe here a novel, isoform-selective activator of the TRPML2 channel and describe how TRPML2 activation enhances endosomal trafficking to induce inflammatory mediator release in LPS-stimulated macrophages. Until now, selective activators for TRPML2 had not been available. In an effort to identify such selective activators we synthesized >80 chemical compounds by systematic variation of the known lead structures SN-2 and SF-51/ML-SA1 (*Grimm et al., 2010*; *Shen et al., 2012*), generating a library of analogues of sufficient size to deduce structure-activity relationships. In the ML-SA1 series, improved TRPML2 activation was achieved by modification of the length of the acyl spacer, but the resulting selective activators were of only intermediate efficacy and potency. By contrast, the activator ML2-SA1 from the series of norbornene-derived isoxazolines (based on SN-2) is characterized by high TRPML2 subtype selectivity as well as high efficacy and potency, rendering this new small molecule a valuable compound for future studies on this ion channel (*Supplementary file 2*). Molecular modeling data support specific binding of ML2-SA1 to the pore region of the channel, as observed for ML-SA1. The binding orientation of ML-SA1 at hTRPML2 was found to be similar to the experimentally observed binding to hTRPML1 (*Schmiege et al., 2017*) which is in agreement with nonselective activation. In contrast, the binding orientation of docked ML2-SA1 at hTRPML1 differs from that found for hTRPML2, suggesting a plausible rationale for its selectivity. In an experimental approach where we investigated the functional consequences of point mutations in hTRPML2 with the endolysosomal patch-clamp technique we found that in mutant G425A activation by ML2-SA1 is selectively lost, while activation by ML-SA1 is preserved, indicating that this amino acid is highly critical for the selective effect of ML2-SA1 on TRPML2.

*Sun et al. (2015)* have recently shown that the levels of TRPML2 are strongly upregulated in macrophages upon TLR4 (toll-like receptor) activation (*Supplementary file 2*). Thus, treatment with LPS was found to lead to TRPML2 upregulation in, for example microglia, peritoneal macrophages, bone marrow derived macrophages, or alveolar macrophages (*Sun et al., 2015*). The authors further found that the translation and secretion of several chemokines such as CCL2 was reduced in TRPML2$^{-/-}$ mice, and concluded that TRPML2 might play a role in the regulation of trafficking and/or

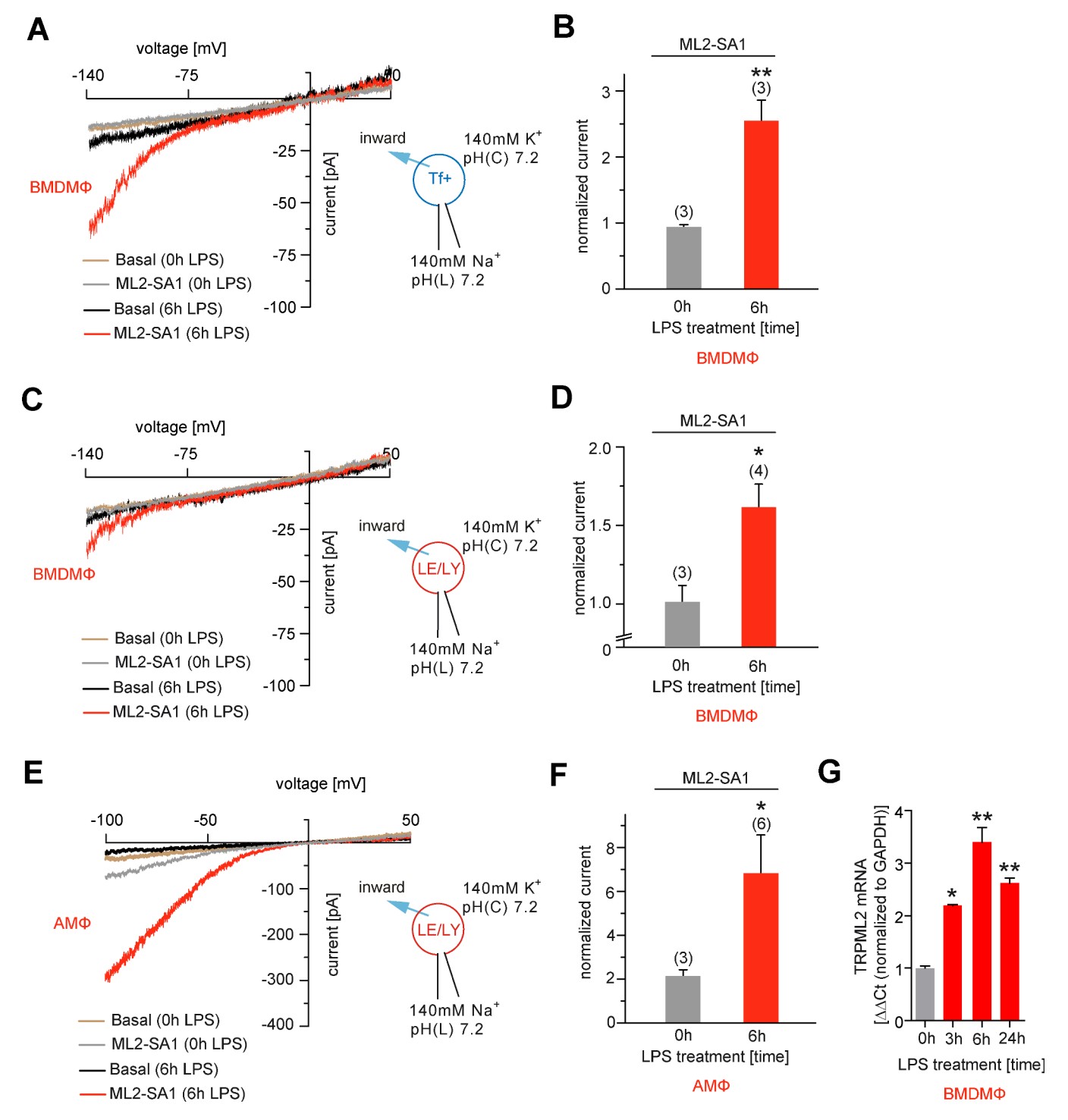

**Figure 4.** Effect of ML2-SA1 on channel currents in endolysosomal organelles isolated from different primary mouse macrophages. (**A**) Representative currents from vacuolin-enlarged/Tf+ vesicles isolated from murine (LPS 6 hr or LPS 0 hr) primary WT BMDMΦ, basal or elicited by an application of 10 μM ML2-SA1. All currents are normalized to basal current without ML2-SA1. (**B**) Statistical summary of data shown in A. (**C**) Representative currents from YM201636-enlarged LE/LY isolated from murine (LPS 6 hr or LPS 0 hr) primary WT bone marrow macrophages (BMDMΦ), basal or elicited by an application of 10 μM ML2-SA1. (**D**) Statistical summary of data shown in C. (**E**) Representative currents from YM201636-enlarged LE/LY isolated from murine (LPS 6 hr or LPS 0 hr) primary WT alveolar macrophages (AMΦ), basal or elicited by an application of 10 μM ML2-SA1. (**F**) Statistical summary of data shown in E. * indicates p<0.05, ** indicates p<0.01, Student's t test, unpaired. (**G**) qPCR data showing levels of TRPML2 expression after 3, 6, and 24 hr LPS treatment compared to untreated (0 hr). * indicates p<0.05, ** indicates p<0.01, one-way ANOVA test followed by Tukey's post-hoc test.

*Figure 4 continued on next page*

*Figure 4 continued*

DOI: https://doi.org/10.7554/eLife.39720.012

secretion of these chemokines. However, it remained unclear whether TRPML2 is directly involved in these processes and whether activation of TRPML2 channel activity would show increased release.

Here, we present data strongly supporting a direct involvement of TRPML2, as direct stimulation of TRPML2 with ML2-SA1 leads to an increase in CCL2 secretion from macrophages. Using the specific TRPML2 agonist and the TRPML2$^{-/-}$ knockout mouse model as control, we demonstrate a positive relationship between TRPML2 activity and CCL2 secretion. Using the endolysosomal patch-clamp technique we demonstrate that TRPML2 is present in LE/LY and EE as well as in Rab11+ and TfR+/Tf+ vesicles (*Supplementary file 2*). However, early endosomes including RE provide more favorable activation conditions for TRPML2 than LE/LY due to their less acidic/neutral luminal pH. In accordance with this, TRPML2 currents elicited with ML2-SA1 in LE/LY isolated from endogenously expressing BMDMΦ were smaller than currents in Tf+ vesicles. In addition, no evidence was found that ML2-SA1 can promote lysosomal exocytosis, while ionomycin or ML-SA1 were able to increase the release of beta-hexosaminidase as previously reported (*Samie et al., 2013*). The subcellular distribution of LAMP1 did also not change during ML2-SA1 treatment and no translocation to the PM was observed. In contrast, ML2-SA1 application was found to significantly promote Tf trafficking through the early/recycling endosomal compartment, arguing for a role of TRPML2 in CCL2 release via the early/recycling endosomal pathway (*Figure 5F*).

Loss of function mutations in the TRPML2-related channel TRPML1 result in lysosomal storage and endolysosomal trafficking defects underlying the neurodegenerative disease mucolipidosis type IV (*Bach, 2001*; *Pryor et al., 2006*; *Chen et al., 2014*). Mechanistically, it was postulated that loss of TRPML1 impairs lysosomal exocytosis (*LaPlante et al., 2006*). It was also suggested that TRPML1 is required for lysosomal pH regulation (*Soyombo et al., 2006*) and for vesicle fusion (*Venkatachalam et al., 2013*) while, very recently, data have been presented, supporting that TRPML1 may regulate lysosomal fission (*Chen et al., 2017a*). A further interesting finding has been presented by *Park et al. (2016)*, suggesting that, in secretory cells, a major role for TRPML1 is to guard against unintended, pathological fusion of lysosomes with other intracellular organelles, for example secretory vesicles. TRPML1 has also been attributed to mediate lysosomal trafficking via Ca$^{2+}$-dependent motor protein recruitment, its activity favoring retrograde lysosomal movement (*Vergarajauregui et al., 2009*; *Li et al., 2016*).

Like TRPML1, TRPML3 was also suggested to regulate membrane trafficking. In particular, it was found to regulate trafficking of early endosomes and to affect endocytosis (*Kim et al., 2009*). *Lelouvier and Puertollano (2011)* further presented data showing that TRPML3 is required for proper calcium homeostasis in the endosomal pathway and that impairment of TRPML3 function leads to defective endosomal acidification and defective membrane trafficking. Surprisingly, the authors found increased endosomal fusion after depletion of TRPML3. Recently, *Miao et al. (2015)* showed that TRPML3 activation, upon neutralization of lysosomal pH, mediates efflux of Ca$^{2+}$ ions from lysosomes, which in turn induces lysosome exocytosis. TRPML3 is normally inactive in highly acidic lysosomes, in contrast to early/recycling endosomes with more neutral pH, but when the pH in the lumen of the lysosome is neutralized, TRPML3 becomes activated, releases Ca$^{2+}$ into the cytosol, which in turn triggers spontaneous exocytosis of the lysosome and its contents.

TRPML2 has been suggested to play a role in the regulation of the Arf6-associated pathway and, more specifically, in the trafficking of GPI-APs (*Karacsonyi et al., 2007*). Arf6 has been implicated in the regulation of endocytosis as well as endocytic recycling and cytoskeleton remodeling. More recently, TRPML2 has been found to increase trafficking efficiency of endocytosed viruses (*Rinkenberger and Schoggins, 2018*). Furthermore, we are showing here that TRPML2 is, like its relatives TRPML1 and 3, Ca$^{2+}$ permeable (*Figure 2—figure supplement 1C*).

Taken together, these findings imply that all three TRPML channels can impact intracellular trafficking processes while the mechanisms how they affect trafficking might differ. While it is likely, based on the available data, that the effect of TRPML2 knockout/activation on CCL2 trafficking and release is occurring at the level of EE/RE, it remains to be further established where along this pathway the effect takes place. Possible scenarios might be: fusion of Golgi vesicles with RE, fission from

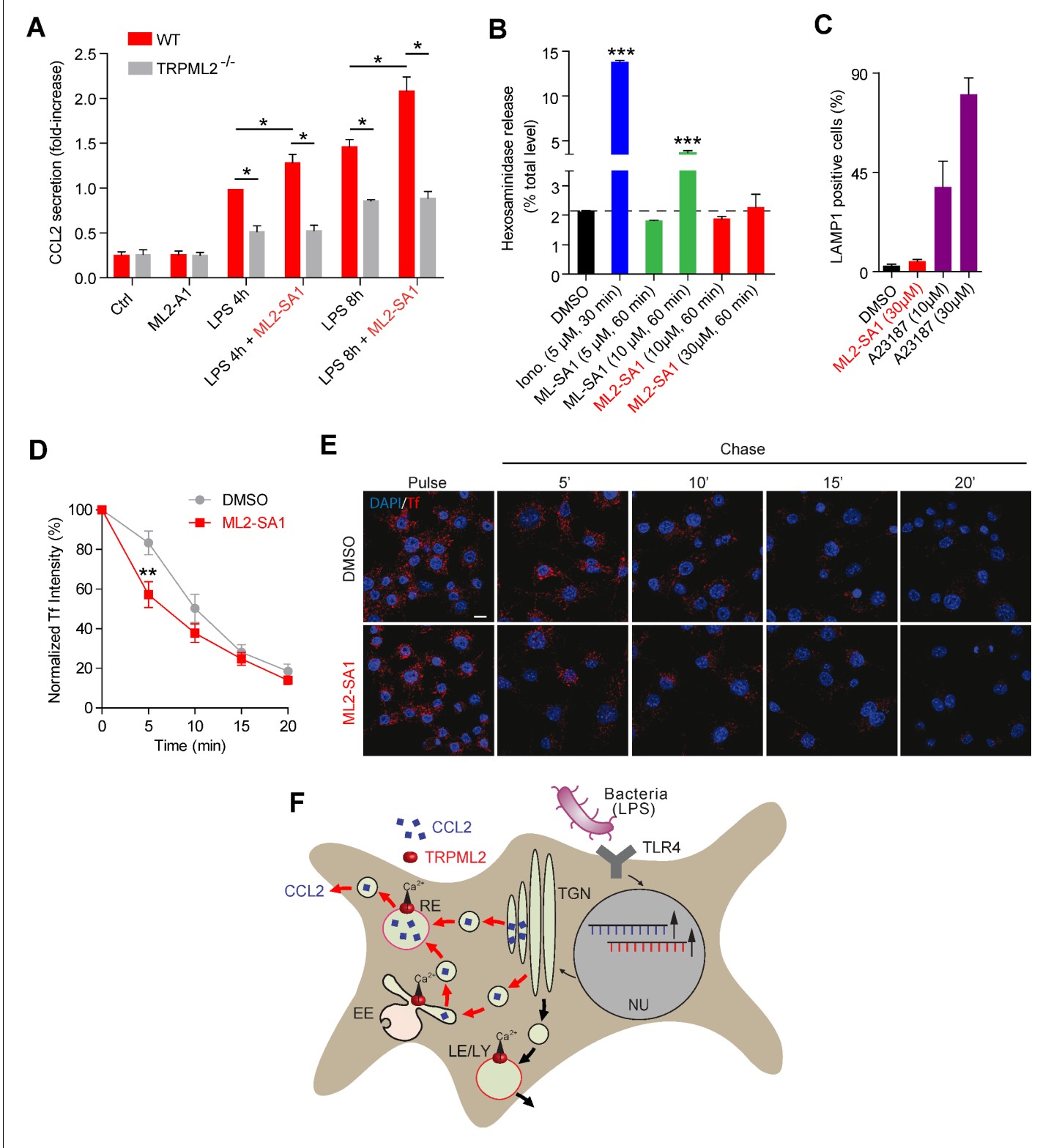

**Figure 5.** Effect of ML2-SA1 on chemokine release from primary mouse macrophages. (**A**) Shown are data obtained from primary WT and TRPML2$^{-/-}$ mouse bone marrow macrophages (BMDMΦ) with and without LPS treatment for 4 hr and 8 hr, respectively. The fraction treated with LPS and 10 μM ML2-SA1 showed significantly increased CCL2 secretion compared to WT controls treated with LPS only. TRPML2$^{-/-}$ cells displayed strongly reduced CCL2 secretion. Shown are normalized mean values ± SEM of 5 mice each. * indicates p<0.05, Student's t test, unpaired. (**B**) Lysosomal exocytosis assay showing the increase in beta-hexosaminidase release upon stimulation with either ionomycin, ML-SA1, or ML2-SA1 (conc. as indicated) from LPS (6 hr)

*Figure 5 continued on next page*

Figure 5 continued

stimulated BMDMΦ. *** indicates p<0.001, one-way ANOVA test followed by Tukey's post-hoc test. (C) Lysosomal exocytosis assay by flow cytometry showing the percentage of cells which show an increase in LAMP1 fluorescence on the plasma membrane. Cells were treated with DMSO, calcium ionophore A23187 (calcimycin), and 30 μM ML2-SA1. (D–E) Recycling endosome assay showing the decrease of Tf mean fluorescence in LPS stimulated RAW264.7 cells, treated with either DMSO or 30 μM ML2-SA1. Scale bar (identical for all images)=10 μm. Plot shows the normalized Tf intensity (shown is the average of 3 independent experiments, each). **p<0.01, two-way ANOVA, repeated measures, followed by Bonferroni post-hoc test. (F) Cartoon showing organelles with functional TRPML2 expression as confirmed by endolysosomal patch-clamp analysis (EE, RE, LE/LY). CCL2 (MCP-1) is hypothesized to be trafficked and secreted via the EE/RE pathway, based on the observation that ML2-SA1 promotes Tf trafficking in the EE/RE compartment, while no effect on lysosomal exocytosis was found. No secretory vesicles are reported to exist in macrophages.
DOI: https://doi.org/10.7554/eLife.39720.013

RE, fusion of Golgi vesicles with EE, fission from EE, fusion of EE-derived vesicles with the RE (*Figure 5F*).

Functionally, we found that ML2-SA1 promotes migration of untreated macrophages towards LPS-treated macrophages. This suggests that TRPML2-dependent CCL2 release is enhancing the inflammatory response by recruiting innate immune cells to the site of inflammation. This is in accordance with the results presented by *Sun et al. (2015)* who found that macrophage migration is impaired in vivo in the absence of TRPML2.

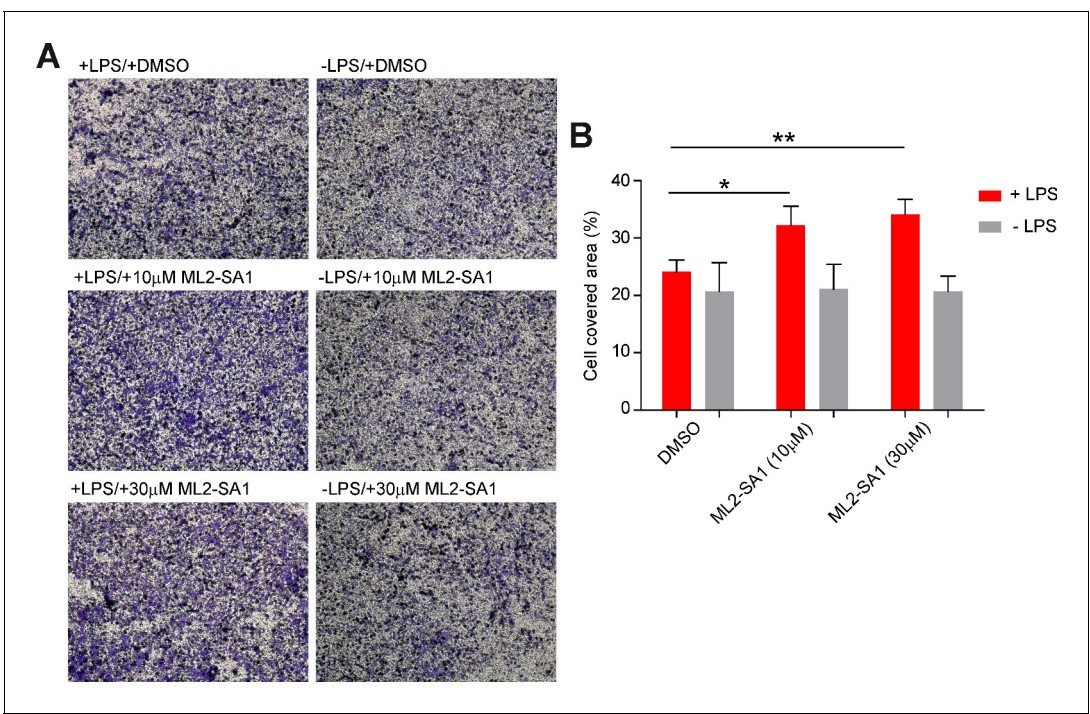

**Figure 6.** Effect of ML2-SA1 on macrophage migration. (**A**) Shown are representative images obtained from a modified Boyden chamber experiment. Images show fixed and crystal violet stained BMDMΦ after 3 hr migration through a transwell chamber along a chemotactic gradient created by BMDMΦ in the lower compartment. Indicated treatments refer to treatment of the cells in the lower compartment. (**B**) Quantification of migration in the modified Boyden chamber setup (**A**) shows a significant increase in migration when LPS pre-treated cells in the lower compartment were subjected to 10 or 30 μM ML2-SA1. Shown are mean values ± SEM of 4 independent experiments. * indicates p<0.05, ** indicates p<0.01, repeated measures, one-way ANOVA with Greenhouse-Geisser correction, followed by Dunnet post-hoc test.
DOI: https://doi.org/10.7554/eLife.39720.014

The following figure supplements are available for figure 6:

**Figure supplement 1.** Modified and classical Boyden chamber setup.
DOI: https://doi.org/10.7554/eLife.39720.015

**Figure supplement 2.** Migration assay without cells in the lower compartment of the classical Boyden chamber.
DOI: https://doi.org/10.7554/eLife.39720.016

CCL2 is known to be a key chemokine regulating migration and infiltration of monocytes/macrophages (*Deshmane et al., 2009*). Since CCL2 is implicated in the pathogenesis of diseases characterized by infiltrates containing macrophages like psoriasis, rheumatoid arthritis, multiple sclerosis, and atherosclerosis (*Deshmane et al., 2009*; *Xia and Sui, 2009*; *Daly and Rollins, 2003*), we postulate that TRPML2 may be an attractive novel target for the treatment of such innate immunity-related inflammatory diseases.

# Materials and methods

## Key resources table

| Designation | Source or reference | Identifiers | Additional information |
|---|---|---|---|
| HEK293 | DSMZ | ACC 305 | |
| HEK 293 stable stably expressing TRPML3-YFP | *Grimm et al. (2010)*; PMID: 20189104 | | |
| HEK 293 stable stably expressing TRPML1-YFP | *Chen et al. (2014)*, PMID: 25119295 | | |
| TRPML1 (encoded by the *Mcoln1* gene) KO mouse; Mcoln1tm1Sasl, C57BL/6 | *Venugopal et al. (2007)*; PMID: 17924347 | MGI ID: 3794204 | |
| TRPML2 (encoded by the *Mcoln2* gene) KO mouse; C57BL/6 | *Sun et al. (2015)*; PMID: 26432893 | MGI: 1915529 | |
| TRPML3 (encoded by the *Mcoln3* gene) KO mouse; Mcoln3tm1. 1Hels, FVB/NJ | *Jörs et al. (2010)*; PMID: 21179200 | MGI ID: 5319089 | |
| anti-LAMP-1 (1D4B) (rat monoclonal) | Santa Cruz | Cat#A-11006; RRID: AB_2134495 | (1:100) |
| Goat anti-Rat IgG (H + L) Secondary Antibody, Alexa Fluor 488 | ThermoFisher | Cat#sc-19992; RRID: AB_2534074 | (1:1000) |
| mcherry-Transferrin Receptor 20 (plasmid) | N/A | Addgene Plasmid #55144 | |
| DsRed-Rab11 (plasmid) | *Choudhury et al. (2002)*; PMID: 12070301 | Addgene Plasmid #12679 | |
| TRPML1-YFP (plasmid) | *Grimm et al. (2010)*, PMID: 20189104 | | |
| TRPML2-YFP (plasmid) | *Grimm et al. (2010)*, PMID: 20189104 | | |
| TRPML3-YFP (plasmid) | *Grimm et al. (2010)*, PMID: 20189104 | | |
| Quikchange primers for TRPML2:YFP A422C | this paper | | forward: CTTCGGTTTTGTTGTTGTG CTGGTATGATTTATCTGGG reverse: CCCAGATAAATCATACCAGC ACAACAACAAAACCGAAG |
| Quikchange primers for TRPML2:YFP A424V | this paper | | forward: CGGTTTTGTGCTTG TGTTGGTATGATTTATCTGGGTTACAC reverse: GTGTAACCCAGATAAATCAT ACCAACACAAGCACAAAACCG |
| Quikchange primers for TRPML2:YFP G425A | this paper | | forward: CGGTTTTGTGCTTGT GCTGCTATGATTTATCTGGGTTACAC reverse: GTGTAACCCAGATAAATCA TAGCAGCACAAGCACAAAACCG |
| Quikchange primers for TRPML2:YFP A453S | this paper | | forward: CTGAACACAGTTTCTG AGTGTCTGTTTTCTCTGG reverse: CCAGAGAAAACAGACA CTCAGAAACTGTGTTCAG |

*Continued on next page*

*Continued*

| Designation | Source or reference | Identifiers | Additional information |
| --- | --- | --- | --- |
| Quikchange primers for TRPML2:YFP V460I | this paper | | forward: TGTCTGTTTTCTCTGATCA ACGGTGATGACATG reverse: CATGTCATCACCGTTGATC AGAGAAAACAGACA |
| Quikchange primers for TRPML2:YFP I498V | this paper | | forward: CCTTCATCAGCCTTTTTATATATA TGGTTCTCAGTCTTTTTATTGC reverse: GCAATAAAAAGACTGAGAACCA TATATATAAAAAGGCTGATGAAGG |
| qPCR Primer for TRPML1 (NM_053177) | www.pga.mgh. harvard.edu/primerbank | PrimerBankID: 16716462 c2 | forward: GCCTTGGGCCAATGGATCA reverse: CCCTTGGATCAATGTCAAAGGTA |
| qPCR Primer for TRPML2 (NM_026656) | this paper | | forward: AATTTGGGGTCACGTCATGC reverse: AGAATCGAGAGACGCCATCG |
| qPCR Primer for TRPML3 (NM_134160) | this paper | | forward: GAGTTACCTGGTGTGGCTGT reverse: TGCTGGTAGTGCTTAATTGTTTCG |
| qPCR Primer for HPRT (NM_013556) | *Hruz et al. (2011)*; PMID: 21418615 | N/A | forward: GCTCGAGATGTCATGAAGGAGAT reverse: AAAGAACTTATAGCCCCCCTTGA |
| Lipopolysaccharides (LPS) from Escherichia coli O26:B6 | Sigma-Aldrich | Cat#L2762 | |
| Lipopolysaccharides (LPS) from Escherichia coli O111:B4 | Sigma-Aldrich | Cat#L4391 | |
| Fura-2, AM, cell permeant | ThermoFisher | Cat#F1201 | |
| Mouse M-CSF, premium grade | Miltenyi Biotech | Cat#130-101-703 | |
| Transferrin from human serum, Alexa FluorTM 546-conjugated | TermoFisher | Cat# T23364 | |
| Transferrin from human serum, Alexa FluorTM 555-conjugated | Thermo Fisher | Cat#T35352 | |
| JE/MCP-1/CCL2 from mouse, recombinant | Sigma-Aldrich | Cat# SRP4207 | |
| YM201636 | Chemdea | Cat#CD0181 | |
| MLSA-1 | Sigma-Aldrich | Cat#SML0627 | |
| PI(3,5)P2 | AG Scientific | Cat#P-1123 | |
| Wortmannin | Sigma-Aldrich | Cat#W1628 | |
| LatrunculinB | Sigma-Aldrich | Cat#L5288 | |
| Vacuolin | Santa Cruz | Cat# sc-216045 | |
| Calcium ionophore A23187 | Sigma-Aldrich | Cat#C7522 | |
| 4-Methylumbelliferyl N-acetyl-b-D-glucosaminide | Sigma-Aldrich | Cat#M2133 | |
| RNeasy Plus Mini Kit | Qiagen | Cat# 74134 | |
| RevertAid first strand cDNA synthesis Kit | ThermoScientific | Cat# K1621 | |
| CD11b MicroBeads, human and mouse | Miltenyi Biotech | Cat#130-049-601 | |
| QuikChange II Site-Directed Mutagenesis Kit | Agilent | Cat#200523 | |
| Mouse/rat CCL2/ JE/MCP-1 Quantikine ELISA Kit | BioLegend | Cat#432707 | |
| Origin8 | OriginLab | | |
| GraphPad Prism | GraphPad Software Inc. | | |

## Endolysosomal patch-clamp and calcium imaging experiments

Whole-LE/LY and whole-EE recordings have been described previously in detail (*Chen et al., 2017a*; *Chen et al., 2017c*). In brief, for whole-LE/LY manual patch-clamp recordings, cells were treated with YM201636 (HEK293 cells: 800 nM o/n; macrophages: 800 nM 1 hr). For whole-EE manual patch-clamp recordings, cells were treated with a combination of 200 nM wortmannin and 10 nM latrunculin B (HEK293 cells: 10–15 min). Cells were treated with compounds at 37°C and 5% $CO_2$. YM201636 was obtained from Chemdea (CD0181), wortmannin and latrunculin B from Sigma (W1628 and L5288), and vacuolin from Santa Cruz (sc-216045). Compounds were washed out before patch-clamp experimentation.

For other organelle patch-clamp recordings, HEK293 cells were transfected with the markers Rab11-DsRed or TfR-mCherry, respectively, and treated with 1 µM vacuolin o/n. Since macrophages could not be transfected with standard transfection protocols or by electroporation, cells were loaded with transferrin-Alexa555 and simultaneously treated with vacuolin for 1 hr to enlarge and visualize vesicles for patch-clamp.

Isolation-micropipettes were used to open up the plasma membrane, and push the enlarged vesicle of interest out of the cell. Afterwards, electrode-micropipettes were applied to patch-clamp the isolated vesicles.

Macrophages were used for experiments within 2–10 days after isolation. Mean capacitance values for Rab11+ vesicles isolated from HEK293 cells was $0.7 \pm 0.2$ (n = 6), for TfR+ vesicles (n = 3) $1.4 \pm 0.3$ pF, for EE (n = 10) $0.4 \pm 0.1$ pF, and for LE/LY (n = 51) $1.0 \pm 0.2$ pF. For LE/LY isolated from primary macrophages it was $0.8 \pm 0.1$ pF (n = 41), for Tf-loaded vesicles $1.3 \pm 0.5$ pF (n = 8). Currents were recorded using an EPC-10 patch-clamp amplifier (HEKA, Lambrecht, Germany) and PatchMaster acquisition software (HEKA). Data were digitized at 40 kHz and filtered at 2.8 kHz. Fast and slow capacitive transients were cancelled by the compensation circuit of the EPC-10 amplifier. Recording glass pipettes were polished and had a resistance of 4–8 MΩ. For all experiments, salt-agar bridges were used to connect the reference Ag-AgCl wire to the bath solution to minimize voltage offsets. Liquid junction potential was corrected. For the application of the lipids (A.G. Scientific) or small molecule agonists (ML2-SA1, ML-SA1), cytoplasmic solution was completely exchanged by cytoplasmic solution containing agonist. Unless otherwise stated, cytoplasmic solution contained 140 mM K-MSA, 5 mM KOH, 4 mM NaCl, 0.39 mM $CaCl_2$, 1 mM EGTA and 10 mM HEPES (pH was adjusted with KOH to 7.2). Luminal solution contained 140 mM Na-MSA, 5 mM K-MSA, 2 mM Ca-MSA 2 mM, 1 mM $CaCl_2$, 10 mM HEPES and 10 mM MES (pH was adjusted with NaOH to 7.2). For optimal conditions of TRPML1, luminal pH was adjusted to 4.6 and Na-MSA was used in the luminal solution. For optimal conditions of TRPML2, luminal pH was adjusted to 7.2 and Na-MSA was used in the luminal solution. For optimal conditions of TRPML3, luminal pH was adjusted to 7.2 and K-MSA was applied to replace Na-MSA in the luminal solution. In all experiments, 500 ms voltage ramps from −100 to +100 mV were applied every 5 s, holding potential at 0 mV. The current amplitudes at −100 mV were extracted from individual ramp current recordings. All statistical analysis was done using Origin8 software.

Calcium imaging experiments were performed using fura-2 as described previously (*Grimm et al., 2012a*). Briefly, HEK293 cells were plated onto glass coverslips, grown over night and transiently transfected with the respected cDNAs using TurboFect transfection reagent (Thermo Scientific). After 24–48 h cells were loaded for 1 hr with the fluorescent indicator fura2-AM (4 µM; Invitrogen) in a standard bath solution (SBS) containing (in mM) 138 NaCl, 6 KCl, 2 $MgCl_2$, 2 $CaCl_2$, 10 HEPES, and 5.5 D-glucose (adjusted to pH 7.4 with NaOH). Cells were washed in SBS for 30 min before measurement. Calcium imaging was performed using a monochromator-based imaging system (Polychrome IV mono-chromator, TILL Photonics).

## Computational methods

Analysis of electron density map. The electron density maps for the cryo-electron microscopy structures of hTRPML1 and hTRPML3 in open agonist-bound form (PDB IDs: 5WJ9 and 6AYF, respectively) were downloaded from the Protein Data Bank (PDB; ww.rcsb.org) (*Berman et al., 2000*) and visualized in PyMOL (The PyMOL Molecular Graphics System, Version 1.7.4 Schrödinger, LLC). Homology modelling of TRPML2. The amino acid sequence of hTRPML2 was retrieved from UniProt

(*The UniProt Consortium, 2017*); Accession number: Q8IZK6-1) and a Blast (*Altschul et al., 1990*) search using BLOSUM62 matrix was performed against the PDB to find the closest homologues. Subsequently, sequence alignment of hTRPML2 to the top scored template, hTRPML3 (Sequence identity 59%), was conducted in MOE2012.10 (*Molecular Operating Environment (MOE)*, 2016.08; Chemical Computing Group Inc., 1010 Sherbooke St. West, Suite #910, Montreal, QC, Canada, H3A 2R7, 2016) and the alignment file was used to generate the homology model using MODELLER 9.11 (*Webb and Sali, 2014*). Ligand-bound homology models of hTRPML2 were finally built using the agonist-bound structure of hTRPML3 (PDB ID: 6AYF) and ranked according to their DOPE score (*Shen and Sali, 2006*). Molecular docking to hTRPML1 and −2. The ligands were prepared for docking using the LigPrep tool as implemented in Schrödinger's software (*Schrödinger Release 2017–1*: LigPrep, Schrödinger, LLC, New York, NY, 2017), where the two stereoisomers of ML2-SA1 were generated and energy minimized using the OPLS force field. Conformers of the prepared ligands were calculated with ConfGen using the default settings and allowing minimization of the output conformations.

Protein preparation. The cryo-electron microscopy structure of the open conformation of hTRPML1 in complex with ML-SA1 (PDB ID: 5WJ9) and the generated hTRPML2 homology model were prepared with Schrödinger's Protein Preparation Wizard (*Schrödinger Release 2017–1*: Schrödinger Suite 2017–1 Protein Preparation Wizard; Epik, Schrödinger, LLC, New York, NY, 2016; Impact, Schrödinger, LLC, New York, NY, 2016; Prime, Schrödinger, LLC, New York, NY, 2017): Hydrogen atoms were added and the H-bond network was subsequently optimized. The protonation states at pH 7.0 were predicted using the PROPKA tool in Schrödinger. The structures were finally subjected to a restrained energy minimization step using the OPLS2005 force field (RMSD of the atom displacement for terminating the minimization was 0.3 Å).

The receptor grid preparation for the docking procedure was carried out by assigning the agonist as the centroid of the grid box. The generated ligand conformers were docked into the proteins using Glide (*Small-Molecule Drug Discovery Suite 2017–1*: Glide, Schrödinger, LLC, New York, NY, 2017) in the Standard Precision mode. A total of 100 poses per ligand conformer were included in the post-docking minimization step and a maximum of 20 docking clusters were output for each ligand. Redocking of the ligand ML-SA1 into the hTRPML1 pocket gave a docking pose with root mean square deviation of 1.22 Å for the top-ranked solution (*Figure 3—figure supplement 1A*).

## Cell culture of primary macrophages isolated from knockout and WT mice

For preparation of primary alveolar macrophages (AMΦ), mice were deeply anesthetized and euthanized by exsanguination. Afterwards, the trachea was carefully exposed and cannulated by inserting a 20 gauge catheter (B. Braun, cat. no. 4252110B). AMΦ were harvested by eight consecutive lung lavages with 1 ml of DPBS each. After a centrifugation step, cells were immediately collected and cultured in RPMI 1640 medium supplemented with 10% fetal bovine serum and 1% antibiotics. AMΦ were directly seeded onto 12 mm glass cover slips and used for experiments within 5 days after preparation. Bone marrow-derived macrophages (BMDMΦ) were isolated from femur and tibias of mice. Thus, bones were isolated and bone marrow was flushed with 10 ml PBS using a sterile 25 gauge needle. Cells were obtained by centrifugation, resuspended and subsequently cultured in 10 cm petri dishes in RPMI 1640 medium supplemented with 10% fetal bovine serum and 1% penicillin/streptomycin and 40 ng/mL murine M-CSF (Miltenyi Biotech). Cells were incubated for 5 days, before they were plated onto poly-L-lysine coated cover slips for experiments. All cells were maintained at 37°C in 5% $CO_2$ atmosphere. If necessary, cells were stimulated with 1 µg/mL LPS (Escherichia coli O26:B6, Sigma, L2762) prior to experiments for different time periods as stated in the text. Animals were used under approved animal protocols and University of Munich (LMU) Institutional Animal Care Guidelines.

## Measurement of CCL2 content in bmdm$\varphi$ culture supernatants by ELISA

Cell culture supernatants from WT or TRPML2$^{-/-}$ BMDMΦ were collected at 4 hr or 8 hr following LPS treatment in the presence or absence of TRPML2 agonist (ML2-SA1), and CCL2 was measured

using an ELISA kit (BioLegend, 432707), per the manufacturer's instructions. Cell culture supernatants were diluted ten times for the assay, and 50 µL diluted supernatant was assessed.

## Transferrin trafficking assay

RAW264.7 cells were seeded overnight with 0.1 mg/mL of lipopolysaccharide (LPS) (L4391, Sigma). Then, cells were loaded for 20 min at 37°C with transferrin from human serum, Alexa Fluor 546-conjugated (T23364, ThermoFisher) at the concentration of 50 µg/mL in complete medium (DMEM 10% FBS). The analysis of recycling kinetics was performed by chasing for 5, 10, 15 and 20 min in complete media plus 50 µg/mL of unconjugated transferrin (T0665, Sigma) in the presence of either DMSO or ML2-SA1 (30 µM). Before fixation with 4% paraformaldehyde (PFA), non-internalized transferrin was acid-stripped (150 mM NaCl, 0.5% acetic acid in $H_2O$) for 30 s. Images were acquired using a Zeiss LSM 800 with 63x magnification.

## Lysosomal exocytosis assay (FACS)

RAW264.7 cells were seeded overnight with 0.1 mg/mL of lipopolysaccharide (LPS) (L4391, Sigma). Then, cells were treated with DMSO, calcium ionophore A23187 (C7522, Sigma) or ML2-SA1 for 3 hr. After 3 h cells were collected and stained with LAMP1 antibody (SC-19992, Santa Cruz) in PBS (1% BSA) during agitation for 20 min (4°C). Cells were then collected by centrifugation and resuspended in PBS (1% BSA) with goat anti-rat, Alexa488 (A-11006 ThermoFisher) during agitation for 1 hr (4°C). Finally, cells were washed in PBS and left on ice until FACS analysis. Cells were loaded into the FACS machine using a nozzle of 100 µm and the LAMP1 fluorescence intensity was measured using a 488 nm excitation laser and a FITCH (530/30 nm) emission filter. The threshold was set using DMSO-treated samples, and 1000 events were counted for each condition.

## Lysosomal exocytosis assay (Hexosaminidase)

For measurement of lysosomal hexosaminidase enzyme release, bone marrow macrophages were treated with ML2-SA1, ML-SA1 or DMSO in serum-free RPMI medium, concentrations and durations as indicated. Ionomycin was used as control. After treatment, supernatants were collected, centrifuged and incubated with natrium citrate buffer (pH 4.5) and 4-Methylumbelliferyl N-acetyl-β-D-glucosaminide (M1233, Sigma, 1 mM final concentration) for 1.5 hr. Cells were lysed with Triton-X buffer and lysates were processed in parallel. The reaction was stopped by adding glycin buffer to the samples and the turnover of hexosaminidase substrate was detected as fluorescence (Exitation: 365 nm; Emission: 450 nm) using a plate reader (Spectramax ID3, Molecular Devices). The increase in substrate turnover was analyzed as fluorescence increase in supernatants relative to lysates.

## Site-directed mutagenesis

Generation of point mutant isoforms of hTRPML2 (encoded by the *MCOLN2* gene) was performed as described previously (*Grimm et al., 2010*) using the QuikChange protocol. The following primers were used to generate hTRPML2 mutant isoforms: A422C forward primer: CTTCGGTTTTGTTGTTGTGCTGGTATGATTTATCTGGG; A422C reverse primer: CCCAGATAAATCATACCAGCACAACAACAAAACCGAAG; A424V forward primer: CGGTTTTGTGCTTGTGTTGGTATGATTTATCTGGGTTACAC; A424V reverse primer: GTGTAACCCAGATAAATCATACCAACACAAGCACAAAACCG; G425A forward primer: CGGTTTTGTGCTTGTGCTGCTATGATTTATCTGGGTTACAC; G425A reverse primer: GTGTAACCCAGATAAATCATAGCAGCACAAGCACAAAACCG; A453S forward primer: CTGAACACAGTTTCTGAGTGTCTGTTTTCTCTGG; A453S reverse primer: CCAGAGAAAACAGACACTCAGAAACTGTGTTCAG; V460I forward primer: TGTCTGTTTTCTCTGATCAACGGTGATGACATG; V460I reverse primer: CATGTCATCACCGTTGATCAGAGAAAACAGACA; I498V forward primer: CCTTCATCAGCCTTTTTATATATATGGTTCTCAGTCTTTTTATTGC; I498V reverse primer: GCAATAAAAAGACTGAGAACCATATATATAAAAAGGCTGATGAAGG.

## Macrophage migration experiments

ML2-SA1 effects on macrophage migration were assessed by a modified Boyden chamber setup (*Figure 6—figure supplement 2*). In the modified Boyden chamber setup, BMDMΦ were plated onto poly-L-lysine coated cover slips in a twenty-four well plate (lower compartment) in the presence or absence of 1 µg/ml LPS for 6 hr. After 6 hr, media was replaced with media containing 10 or 30

μM ML2-SA1 or DMSO. $1 \times 10^5$ BMDMΦ were placed on top of the transwell chamber (Corning) in media without any compound. Transwell chambers were placed into the twenty-four well plate and incubated for 3 hr at 37°C in 5% $CO_2$ atmosphere. In the classical Boyden chamber approach a twenty-four well plate was filled with media containing either DMSO, 1 μg/ml LPS and DMSO, 1 μg/ml LPS and 30 μM ML2-SA1, or 10 ng/ml CCL2. Transwell chambers were equally prepared and incubated. Migrated cells were fixed and stained with crystal violet/methanol. The top of the transwell chamber was cleaned an images were taken. Cell covered area was determined with ImageJ (NIH, Bethesda, MD).

## Acknowledgement

We thank Lars Allmendinger, Yu-Kai Chao, Berit Noack, Martina Stadler, Wolfgang Wilfert, and Christopher Wolf for technical support. This work was supported, in part, by funding of the German Research Foundation (SFB/TRR152 projects P04 to CG, P06 to CW-S., and P12 to MB. as well as SFB1123 project B1 to LMH and DT), the NCL (Neuronal Ceroid Lipofuscinosis) Foundation Award 2016 to CG, and the University of Pennsylvania Orphan Disease Center and the Mucolipidosis IV Foundation Grant MDBR-17–120 ML4 to CG.

## Additional information

### Funding

| Funder | Grant reference number | Author |
|---|---|---|
| Deutsche Forschungsge-meinschaft | SFB/TRR152 P04 | Christian Grimm |
| Mucolipidosis IV Foundation | MDBR-17-120-ML4 | Christian Grimm |
| Deutsche Forschungsge-meinschaft | SFB/TRR152 P06 | Christian Wahl-Schott |
| Deutsche Forschungsge-meinschaft | SFB/TRR152 P12 | Martin Biel |
| Deutsche Forschungsge-meinschaft | SFB1123 B1 | Lesca M Holdt Daniel Teupser |

The funders had no role in study design, data collection and interpretation, or the decision to submit the work for publication.

### Author contributions

Eva Plesch, Data curation, Formal analysis, Methodology; Cheng-Chang Chen, Conceptualization, Data curation, Formal analysis, Methodology, Writing—review and editing; Elisabeth Butz, Anna Scotto Rosato, Einar K Krogsaeter, Rosa Puertollano, Conceptualization, Data curation, Formal analysis, Methodology; Hua Yinan, Karin Bartel, Conceptualization, Data curation, Formal analysis, Investigation; Marco Keller, Supervision, Writing—review and editing; Dina Robaa, Data curation, Formal analysis; Daniel Teupser, Angelika M Vollmar, Resources, Supervision, Funding acquisition; Lesca M Holdt, Resources, Supervision, Funding acquisition, Methodology; Wolfgang Sippl, Resources, Data curation, Formal analysis, Supervision, Methodology; Diego Medina, Conceptualization, Resources, Supervision; Martin Biel, Resources, Funding acquisition; Christian Wahl-Schott, Resources, Funding acquisition, Methodology; Franz Bracher, Conceptualization, Resources, Data curation, Formal analysis, Supervision, Methodology; Christian Grimm, Conceptualization, Resources, Data curation, Formal analysis, Supervision, Funding acquisition, Validation, Investigation, Visualization, Methodology, Writing—original draft, Project administration, Writing—review and editing

### Author ORCIDs

Elisabeth Butz http://orcid.org/0000-0002-5956-6434
Einar K Krogsaeter http://orcid.org/0000-0001-8232-5498

Rosa Puertollano [ID] http://orcid.org/0000-0002-1106-5489
Christian Grimm [ID] http://orcid.org/0000-0002-0177-5559

## Ethics

Animal experimentation: This study was performed where applicable in strict accordance with the recommendations in the Guide for the Care and Use of Laboratory Animals of the National Institutes of Health. This study was performed where applicable in strict accordance with the recommendations of the Bavarian Government (ROB; AZ_55.2-1-54-2532-27-2015).

## Decision letter and Author response

Decision letter https://doi.org/10.7554/eLife.39720.021
Author response https://doi.org/10.7554/eLife.39720.022

## Additional files

### Supplementary files

• Supplementary file 1. Synthesis details and analytical data
DOI: https://doi.org/10.7554/eLife.39720.017

• Supplementary file 2. Summary of characteristics of TRPML channels
DOI: https://doi.org/10.7554/eLife.39720.018

• Transparent reporting form
DOI: https://doi.org/10.7554/eLife.39720.019

### Data availability

All data generated or analysed during this study are included in the manuscript and supporting files.

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
