## [Decision Letter]

Thank you for submitting your article "Selective agonist of TRPML2 reveals direct role in chemokine release from innate immune cells" for consideration by *eLife*. Your article has been reviewed by two peer reviewers, and the evaluation has been overseen by a Reviewing Editor and Richard Aldrich as the Senior Editor. The following individual involved in review of your submission has agreed to reveal his identity: Antony Galione (Reviewer #2).

The reviewers have discussed the reviews with one another and the Reviewing Editor has drafted this decision to help you prepare a revised submission.

Summary:

TRPML channels are expressed throughout the endolysosomal system and in some cases at the plasma membrane. Of the three isoforms of TRPMLs, TRPML2 is the least well studied. In this manuscript the authors have developed a selective TRPML2 agonist and used this in concert with macrophages from TRPML2^-/-^ mice to demonstrate a functional role for TRPML2 channels in macrophage physiology. The generation of the agonist is an impressive tour de force and only relatively minor revisions are needed regarding the mechanism of effect on chemokine release.

Essential revisions:

1) Regarding the agonist action on the channel. Can you provide additional data on the time course of the agonist effect on TRPML2 currents? It would be ideal to have this data in a new main figure panel.

2) The Discussion should be amplified or qualified regarding the mechanism for "direct" events that lead to chemokine secretion and chemotaxis. The question is: how, precisely, does the channel play a role in secretion? You are encouraged to provide a scheme and to discuss what is clearly established and what needs further work. How do you interpret the increase in currents induced by the agonist? Is there a change in ion selectivity or open probability? A change in gating mode? Can you discuss how they think TRPML2 activation leads (directly) to secretion? Is it via cytosolic calcium? What are the effectors? Can you elaborate a bit more on possible mechanisms?

3) Regarding ML2-SA1 effects: Please elaborate a bit more on the potentiation of the LPS effects with ML2-SA1. What is known about endogenous activation of these channels? The Boyden migration assays show a small enhancement + ML2-SA1 in the bar chart. The effects look larger in the images. Please comment. Please clarify the relationship between CCL2 release and migration. The relationship appears complex and non-linear. Does ML2-SA1 affect TPCs, the other major cation channel family of the endolysosomal system? More discussion is needed concerning how CCL2 is released from cells. The authors have ruled out lysosomal exocytosis, but more discussion of alternative mechanisms (or even experiments) would be helpful and important.

---

## [Author Response]

Essential revisions:1) Regarding the agonist action on the channel. Can you provide additional data on the time course of the agonist effect on TRPML2 currents? It would be ideal to have this data in a new main figure panel.

We now provide data on the time course of the ML2-SA1 effect on TRPML2 currents as requested (see new Figure 2G).

2) The Discussion should be amplified or qualified regarding the mechanism for "direct" events that lead to chemokine secretion and chemotaxis. The question is: how, precisely, does the channel play a role in secretion? You are encouraged to provide a scheme and to discuss what is clearly established and what needs further work.

We agree with the reviewers and did recognize that the Discussion section needs to be improved, in particular relevant literature in the field is now discussed in more detail alongside suggested effects of TRPML channel activation on endolysosomal trafficking, secretion, fusion/fission and endocytosis. We have added the following section to the Discussion:

“Loss of function mutations in the TRPML2-related channel TRPML1 result in lysosomal storage and endolysosomal trafficking defects underlying the neurodegenerative disease mucolipidosis type IV (Bach, 2001; Pryor et al., 2006; Chen et al., 2014). […] Possible scenarios might be: fusion of Golgi vesicles with RE, fission from RE, fusion of Golgi vesicles with EE, fission from EE, fusion of EE-derived vesicles with the RE.”

To better visualize these possibilities we now provide a new scheme (see new Figure 5F), showing in more detail what the possible release pathways are that TRPML2 might be involved in and what consequences would result from TRPML2 KO or direct activation. It is established that:

TRPML2 is functionally active in LE/LY, EE and TfR+/Rab11+ RE. In BMDMs there is less TRPML2 channel activity in LE/LY than in RE. It is unclear whether TRPML2 is expressed in Golgi-derived vesicles or in vesicles which are derived from EE or RE by fission processes. According to literature there are no secretory vesicles present in macrophages.

CCL2 release/secretion from BMDMs is increased when applying ML2-SA1 and decreased in TRPML2KO cells, suggesting that release depends, at least in part, on TRPML2 activity.

Transferrin trafficking through the EE/RE system is accelerated by ML2-SA1 while lysosomal exocytosis is not affected.

What remains speculative is the exact location of interference with the CCL2 trafficking process and where along the pathway the effect might take place: possible scenarios are, as now pointed out in the Discussion, fusion of Golgi vesicles with RE, fission from RE, fusion of Golgi vesicles with EE, fission from EE, fusion of EE derived vesicles with the RE.

How do you interpret the increase in currents induced by the agonist? Is there a change in ion selectivity or open probability? A change in gating mode?

Using the cryo-EM technique, the structure of TRPML1 with ML-SA1 has been solved recently (Schmiege et al., 2017) as well as the structure of TRPML3 with ML-SA1 (Zhou et al., 2017). Schmiege et al. demonstrated that ML-SA1 induces a conformational change of pore helix 1 and transmembrane domain 6 (TM6), effectively dilating the lower selectivity filter and lower hydrophobic gate in a manner reminiscent of capsaicin binding TRPV1. Similarly for TRPML3, Zhou et al. demonstrated ML-SA1 binding to induce a conformational change of the pore loop alongside a TM6 rotation, dilating the pore sufficiently to conduct both hydrated monovalent and divalent cations. We have shown that ML2-SA1 very likely binds in the same binding pocket as ML-SA1 (Figure 3). We therefore speculate that ML2-SA1 may also lead to a pore dilation and thus current increase.

Can you discuss how they think TRPML2 activation leads (directly) to secretion? Is it via cytosolic calcium? What are the effectors? Can you elaborate a bit more on possible mechanisms?

See also answer above (2). Calcium has been postulated to be an important trigger of intracellular vesicle fusion and fission processes as well as fusion with the plasma membrane. We have described above the different locations along the endolysosomal pathway where TRPML2 is functionally expressed (patch clamp data) and where and how it might interfere with the different trafficking steps. We hypothesize that it is the calcium release from the organelle into the cytosol which is the relevant process controlled by TRPML2. ML2-SA1 has been shown to increase cytosolic calcium in calcium imaging experiments. To demonstrate direct calcium release via TRPML2 we now provide endolysosomal patch-clamp experiments assessing the calcium permeability of TRPML2 (see new Figure 2—figure supplement 1C).

3) Regarding ML2-SA1 effects: Please elaborate a bit more on the potentiation of the LPS effects with ML2-SA1.

LPS promotes CCL2 secretion as shown in Figure 5A. The effect on CCL2 release mediated by applying LPS alone is partially dependent on TRPML2 as TRPML2KO leads to a significant reduction in CCL2 release. LPS upregulates the expression of TRPML2 (Sun et al., 2015), hence more TRPML2 would become active (stimulated by endogenous ligands present) or available for activation (e.g. through ML2-SA1) after upregulation of TRPML2 expression. It is also likely that ML2-SA1 increases open-probability and/or pore dilation, thus further increasing TRPML2 activity.

What is known about endogenous activation of these channels?

Generally, TRPML channels can be activated by their putative endogenous ligand PI(3,5)P_2_. PI(3,5)P_2_ occurs in LE/LY membranes and, to a lesser extent, in early endosomal membranes. Furthermore, TRPML2 is pH regulated (Figure 2—figure supplement 1; low pH decreases channel activity). How and under which circumstances endogenous ligand(s) become available for TRPML2 activation remains unclear.

The Boyden migration assays show a small enhancement + ML2-SA1 in the bar chart. The effects look larger in the images. Please comment.

We reassessed our statistical analysis. We confirm the validity of the statistical analysis but do now present a selection of alternative images, better reflecting the results of the statistical analysis (see new Figure 6 and new Figure 6—figure supplement 2).

Please clarify the relationship between CCL2 release and migration. The relationship appears complex and non-linear.

When we incubate BMDM with CCL2 (10 ng/ml), we see an increase in migration from 20% (control) to 40% (Figure 6—figure supplement 2). When we incubate with ML2-SA1, we see an increase in migration from 20/25% (control) to 35%. We only see an effect in LPS treated cells. In fact, -LPS BMDM migrate in the direction of +LPS BMDM when both are exposed to ML2-SA1 (see scheme, Figure 6—figure supplement 1). We have not performed multiple dose-response measurements. However, the experiments as presented show a clear effect of ML2-SA1 on cell migration, allowing us to conclude that ML2-SA1 can mimic the effect of CCL2.

Does ML2-SA1 affect TPCs, the other major cation channel family of the endolysosomal system?

We now provide endolysosomal patch-clamp data showing that ML2-SA1 has no effect on TPC1 and TPC2 (see new Figure 2—figure supplement 3C-F).

More discussion is needed concerning how CCL2 is released from cells. The authors have ruled out lysosomal exocytosis, but more discussion of alternative mechanisms (or even experiments) would be helpful and important.

We now discuss in more detail possible secretion routes of CCL2 from macrophages. See also answer above (2).